# Genome-Wide Identification and Characterization of Diterpenoid Pathway CYPs in *Andrographis paniculata* and Analysis of Their Expression Patterns under Low Temperature Stress

**DOI:** 10.3390/ijms251910741

**Published:** 2024-10-05

**Authors:** Mingyang Sun, Jingyu Li, Shiqiang Xu, Yan Gu, Jihua Wang

**Affiliations:** 1Guangdong Provincial Key Laboratory of Crop Genetic Improvement, Crop Research Institute, Guangdong Academy of Agricultural Sciences, Guangzhou 510640, China; sunmingyangphy@163.com (M.S.); jingyuli138@gmail.com (J.L.); shiqxu0376@163.com (S.X.); guyan@gdaas.cn (Y.G.); 2Guangdong Provincial Engineering & Technology Research Center for Conservation and Utilization of the Genuine Southern Medicinal Resources, Guangzhou 510640, China

**Keywords:** *Andrographis paniculata*, diterpenoids, LT stress, CYPs, functional annotation

## Abstract

*Andrographis paniculata* is known for its diterpenoid medicinal compounds with antibacterial and anti-inflammatory properties. However, it faces production and cultivation challenges due to low temperatures (LTs). Cytochrome P450 monooxygenases (CYPs) are key enzymes in diterpenoid accumulation. Nevertheless, the functions and LT-related expression patterns of diterpenoid pathway *CYPs* in *Andrographis paniculata* remain poorly understood. In this study, 346 CYPs were discovered in *Andrographis paniculata*. Among them, 328 CYPs belonged to 42 known subfamilies. The remaining 17 CYPs might have represented novel subfamilies unique to this species. A total of 65 candidate CYPs associated with diterpenoid modification were identified. Of these, 50 were transmembrane proteins, and 57 were localized to chloroplasts. The CYP71 subfamily was the most abundant and had the highest motif diversity. Promoters of all candidate *CYPs* commonly contained elements responsive to gibberellins (GAs), methyl jasmonate (MeJA), and abiotic stresses. Notably, the XP_051152769 protein, corresponding to a *CYP* gene over 40,000 bp in length, featured an extraordinarily long intron (40,751 nts). Functional elements within this intron were related to LT, GAs, and dehydration pathways. Based on the promoter element arrangement and subfamily classification, 10 representative candidate CYPs were selected. Under LT stress, significant expression changes were observed in three representative *CYPs*: *CYP71D*, *ent-kaurenoic acid oxidase* (*KAO*), and *ent*-*kaurene oxidase* (*KO*). *KAO* and *KO* were significantly upregulated during early LT stress. KAO and KO interacted with each other and jointly interacted with GA20OX2-like. *CYP71D* acted as a negative response factor to LT stress. Among the 37 proteins interacting with CYP71D, 95% were CYPs. This study provides a critical preliminary foundation for investigating the functions of diterpenoid pathway CYPs in *Andrographis paniculata*, thereby facilitating the development of LT-tolerant cultivars.

## 1. Introduction

*Andrographis paniculata* is a common medicinal plant belonging to the *Acanthaceae* family within the *Lamiales* order. Native to the tropical regions of India and Southeast Asia, *Andrographis paniculata* is extensively cultivated in the Guangdong, Fujian, and Hainan provinces of China [1]. The primary active component, andrographolide (C_20_H_30_O_5_), is renowned for its significant anti-inflammatory and antipyretic properties [2]. Andrographolide is a labdane-type diterpenoid biosynthesized from the precursor geranylgeranyl pyrophosphate (GGPP). GGPP is produced via two pathways: the cytoplasmic mevalonate (MVA) pathway and the plastidial 2-C-methyl-D-erythritol-4-phosphate (MEP) pathway [3]. The subsequent synthesis of diterpenoids is predominantly catalyzed by enzymes from the Cytochrome P450 (CYP450) family. The CYP450 family includes CYPs and Cytochrome P450 reductases (CPRs) [4]. CYPs enhance the hydrophilicity of substrates by introducing an oxygen atom. CPRs are crucial for the reductive activation of CYPs by transferring electrons from NADPH to CYPs, serving as the rate-limiting enzymes. CYPs perform various functions, such as hydroxylation, epoxidation, deamination, dehydrogenation, dealkylation, and oxidative C-C bond cleavage [5].

In 2011, Nelson and Werck-Reichhart categorized the CYP family based on amino acid sequence similarities. Sequences with over 40% similarity are grouped into the same family, and those with over 55% similarity are grouped into the same subfamily [6]. Plant CYPs encompass 11 families, including seven single-branch families (CYP51, CYP74, CYP97, CYP710, CYP711, CYP727, and CYP746) and four multi-branch families (CYP71, CYP72, CYP85, and CYP86) [7]. Notably, members of the CYP71 family in terrestrial plants are essential in the modification of diterpenoids. These include the subfamilies CYP71, CYP76AH, CYP76AK, CYP76BK, CYP99A, CYP701A (KO), and CYP726A [8]. Within the CYP71D subfamily, representatives have been identified in *Fontainea* as excellent candidates for diterpenoid synthesis [9]. CYP76AH, CYP76AK, and CYP76BK are specific to the *Lamiaceae* family [10]. CYP99A has been identified only in rice (*Oryza sativa*) [11]. CYP726A is unique to the *Euphorbiaceae* family [12]. Members of the CYP72 family’s CYP714 subfamily and the CYP85 family’s CYP88A (KAO), CYP720B, and CYP725A subfamilies also contribute to diterpenoid modification [8]. CYP720B and CYP725A are exclusive to gymnosperms [13]. Subfamilies KO, CYP714, and KAO participate in the catalytic modification of the diterpenoid plant hormones GAs [14]. GAs are crucial endogenous hormones that promote plant growth and development. GGPP undergoes two catalytic steps to form *ent*-kaurene, which KO enzymatically converts to *ent*-kaurenoic acid, subsequently transformed into GA_12_ by KAO [15]. Members of the CYP714 subfamily catalyze the inactivation of GAs by targeting the C-13 position or adjacent carbons of GAs and/or *ent*-kaurenoic acid [16]. Consequently, the significance of the diterpenoid pathway CYPs in plants is clearly evident.

LTs severely restrict plant growth and yield globally, limiting geographic distribution and posing threats to agricultural development [17]. The results of this study indicated that andrographolide was primarily stored in the leaves. Its concentration significantly decreased when the leaves were exposed to prolonged LT. This exposure caused the leaves to change color from green to red. Therefore, analyzing the response of diterpenoid pathway CYPs in *Andrographis paniculata* to LT stress is vital for enhancing its yield and environmental adaptability. This study identified and classified the diterpenoid pathway CYPs in *Andrographis paniculata*. Subsequently, functional annotation and gene expression profiling under LT stress were performed. It establishes an essential molecular foundation for investigating the pathways of andrographolide and GAs in *Andrographis paniculata*.

## 2. Results

### 2.1. Identification and Classification of Total CYPs in Andrographis Paniculata

Based on the plant CYP HMM model, NCBI, and Interpro annotations, 346 CYPs in *Andrographis paniculata* were identified. These were analyzed alongside 92 terrestrial plant diterpenoid pathway CYPs (excluding gymnosperm-specific subfamilies) [8] and 230 Arabidopsis (*Arabidopsis thaliana*) CYPs, totaling 668 proteins, for phylogenetic relationships (Appendix A). The MUSCLE alignment excluded two excessively short proteins (IDs in 4.2.), leaving 666 proteins for further analysis. Their conserved sequences were used to construct the phylogenetic tree. According to Figure 1 and Table 1, the 666 CYPs clustered into 50 subfamilies. Among these, 328 CYPs from *Andrographis paniculata* were assigned to 42 known subfamilies. The remaining 17 CYPs did not cluster into any recognized subfamily and were annotated as ‘CYP family’. Subfamilies CYP702 and CYP708 are unique to Arabidopsis [18], while subfamilies CYP736 and CYP749 contain only *Andrographis paniculata* CYPs. Diterpenoid pathway CYPs originated from the CYP71 family’s CYP71, CYP76, CYP99, and CYP701A subfamilies, the CYP72 family’s CYP714 subfamily, and the CYP85 family’s CYP88A subfamily. The CYP99 subfamily has been identified only in monocots [19]. In the dicot *Andrographis paniculata*, six CYPs were annotated as CYP99 but clustered within the CYP71 subfamily. These CYPs were reassigned to the CYP71 subfamily (Figure 1 and Appendix A). A total of 70 *Andrographis paniculata* CYPs that clustered with terrestrial plant diterpenoid pathway CYPs (marked in the red clade in Figure 1) were selected as candidates.

### 2.2. Identification of Diterpenoid Pathway CYPs in Andrographis Paniculata

Phylogenetic analysis was conducted on 70 candidate CYPs from *Andrographis paniculata* alongside 92 terrestrial plant diterpenoid pathway CYPs using full-length protein sequences (Figure 2). XP_051128741 did not cluster with any subfamily and was excluded. XP_051150854 clustered with the CYP76AH, CYP76AK, and CYP76BK subfamilies, which are specifically expanded in the *Lamiaceae* of *Lamiales*. This suggested that these subfamilies might have a catalytic function unique to *Lamiales* rather than being restricted to *Lamiaceae*. Plant CYPs typically contain the I-helix, K-helix, and a critical domain near the C-terminus, FXXGXRXCXG [20]. In *Andrographis paniculata*, all 345 identified CYPs contained the I-helix and K-helix, but some lacked the complete FXXGXRXCXG. Based on the FXXGXRXCXG motif, 22 sequences were excluded (IDs: XP_051123363, XP_051127248, XP_051128741, XP_051133618, XP_051136989, XP_051140631, XP_051152697, XP_051123609, XP_051125649, XP_051126314, XP_051150718, XP_051131936, XP_051131945, XP_051133611, XP_051137552, XP_051141267, XP_051141281, XP_051146792, XP_051147073, XP_051149418, XP_051149781, and XP_051150050), leaving 323 high-confidence CYPs (Figure 3). Thus, the 70 candidate diterpenoid pathway CYPs were reduced to 65.

### 2.3. Analysis of Basic Information for Diterpenoid Pathway CYPs

Based on database annotations and phylogenetic classification, the symbols of 65 candidate diterpenoid pathway CYPs were revised (Appendix A). Except for XP_051133612 (267 amino acids, aa) and XP_051136915 (620 aa), the lengths of the remaining 63 candidate CYPs were evenly distributed around 500 aa, with an average length of 503 aa. The two members of the CYP88A subfamily, four members of the CYP714 subfamily, and three members of the CYP76 subfamily were localized to the chloroplasts. The sole member of the CYP701A subfamily, XP_051135400 (KO), was localized to the plasma membrane. Of the 55 CYP71 subfamily members, 48 were localized to the chloroplasts. XP_051121925, XP_051133624, and XP_051121134 were localized to the plasma membrane. XP_051126315 and XP_051133729 were found in the vacuolar membrane, while XP_051133853 and XP_051133854 are localized to the endoplasmic reticulum. These results indicated that the 65 candidate CYPs primarily catalyze products of the MEP pathway. A total of 50 candidate CYPs were transmembrane proteins, with 47 CYPs having a transmembrane domain within the first 30 aa. The remaining three candidate CYPs had two transmembrane domains. However, 57 candidate CYPs lacked signal peptides, while the remaining eight were predicted to have signal peptide sequences with a probability of only 51–69%.

### 2.4. Analysis of Feature Information for Diterpenoid Pathway CYPs

Motif distribution within candidate CYPs was subfamily-specific, elucidating systematic evolutionary relationships (Figure 4). CYPs in the three GA pathway subfamilies exhibited fewer motifs compared to those in the CYP76 and CYP71 subfamilies. The CYP714 subfamily contained only motifs 4 and 6. The CYP88A subfamily included motifs 2, 4, 6, and 7. The CYP701A subfamily had an additional motif 5 compared to CYP88A. The CYP76 subfamily conservatively included motifs 1, 2, 4, 5, 6, and 7. Within the CYP71 subfamily, all members, except for the significantly truncated XP_051133612, consistently contained motifs 1, 2, 3, 4, 5, 6, 7, and 10.

*Cis*-acting elements in promoters determine the biological function of genes. According to Figure 4 and Table 2, GA-responsive elements were not limited to *CYP88A*, *CYP701A*, and *CYP714* subfamilies; *CYPs* in the *CYP76* and *CYP71* subfamilies could also respond to the GA pathway. Additionally, candidate *CYPs* generally responded to abscisic acid (ABA) and anaerobic induction. ABA is involved in stress response, while anaerobic induction is associated with flooding damage. Except for one *CYP* in the *CYP701A* subfamily, all other subfamilies had *CYPs* participating in auxin signaling and environmental stresses such as LT. MeJA, related to damage, induces chemical defense. In addition to two *CYPs* in the *CYP88A* subfamily, the other four subfamilies included 49 candidate *CYPs* with promoters containing MeJA-responsive elements. Two *CYP71* subfamily *CYPs* (protein IDs: XP_051121134 and XP_051133624) had the highest number of MeJA-responsive elements, each with seven. Another two *CYP71* subfamily candidate *CYPs* (protein IDs: XP_051133617 and XP_051146596) had promoters containing flavonoid biosynthesis elements.

Research on wheat indicates that gene function is determined not only by exons and non-coding regions but also by introns, which significantly affect protein efficacy [21]. Our structural analysis showed that one candidate *CYP* in the *CYP71* subfamily (protein ID: XP_051152769) contained an unusually long intron of 40,751 bp. According to Appendix A, this long intron included multiple functionally decisive *cis*-acting elements involved in responses to light, LT, GA signaling, dehydration, and phosphate starvation. The lengths of other candidate *CYPs* were all within 10,000 bp.

### 2.5. Structural Analysis of Representative Diterpenoid Pathway CYPs

The promoter is divided into core, proximal, and distal regions. The core promoter contains the transcriptional start site (TSS), while the proximal promoter is located near the TSS. Both are primary binding areas for specific transcription factors (TFs). Conversely, the influence of TFs bound to the distal promoter is relatively weaker. In this study, the region located at the upstream one-third of the promoter from the start codon was designated as the core and proximal promoter. We selected 10 representative candidate *CYPs* from each subfamily whose *cis*-acting elements were predominantly located within the core and proximal promoters. These were marked by white circles in Figure 4. The protein IDs of the 10 representative candidate *CYPs* were as follows: XP_051121709, XP_051151156, and XP_051151369 of the CYP714 subfamily; XP_051123242 and XP_051144151 of the CYP88A subfamily; XP_051135400 of the CYP701A subfamily; XP_051150854 of the CYP76 subfamily; and XP_051136915 (CYP71B), XP_051137327 (CYP71D), and XP_051133994 (CYP99) of the CYP71 subfamily. The spatial structures of representative candidate CYPs in the GA pathway subfamilies were relatively conserved. Within the CYP71 subfamily, structural discrepancies existed between CYP71D and the other three members, indicating functional specificity (Figure 5).

### 2.6. Protein Interactions of LT-Responsive Representative CYPs

Andrographolide accumulated in the leaves (Figure 6A). Under LT conditions, the leaves of *Andrographis paniculata* turned red, with andrographolide content significantly decreasing (Figure 6B). To investigate the LT response of representative *CYPs*, *Andrographis paniculata* seedlings were subjected to different LT treatments: room temperature (green leaves, CK), mild stress (slight red leaves, SR), moderate stress (medium red leaves, MR), and severe stress (all red leaves, AR). Gene expression profiling revealed that three representative *CYPs* [protein IDs: XP_051123242 (KAO), XP_051135400 (KO), and XP_051137327 (CYP71D)] showed significant expression changes under LT conditions. These CYPs interacted with other differentially expressed genes (Figure 7). CYP71D had the most interacting proteins (37 proteins). Among them, 35 were CYPs (highlighted in yellow background), and 16 were diterpenoid pathway CYPs (marked with red circles). In the GA pathway, KAO and KO interacted with each other, and both also interacted with XP_051141106 (GA20OX2-like). Additionally, KAO and KO each independently interacted with two other GA pathway proteins (highlighted in blue background). KAO also interacted with two Cytochrome P450 oxidoreductases (CYPOR-like) (Figure 7).

### 2.7. LT Expression Profiling of KO, KAO, and CYP71D-Interacting Genes

The accumulations of *KAO* and *KO* were significantly upregulated under mild LT stress but decreased under moderate to severe LT stress (Figure 8). The LT expression pattern of *KAO* was opposite to that of its interacting partners, *GA20OX1-like* and *GAox-like*. Similarly, *KO* exhibited a negative correlation with *GA2ox-like*, which interacted with it under the LT condition. The LT-related expression of *GA20OX2-like* did not show a clear trend relative to *KAO* and *KO*. The expression of *CYP71D* was significantly reduced under all LT treatments, indicating it was a negative regulator of the LT response. Conversely, transcripts of 25 CYPs interacting with CYP71D accumulated under varying degrees of LT stress. These CYPs served as positive response factors at the corresponding stages.

## 3. Discussion

CYPs play a vital role in diterpenoid metabolism in plants [22]. In this study, we identified and classified diterpenoid-modifying CYPs in *Andrographis paniculata*. A total of 345 CYPs were characterized across the genome, with 328 belonging to 42 known subfamilies. Seventeen CYPs were identified as unreported members of subfamilies specifically expanded in *Andrographis paniculata*. Further characterization of their catalytic functions is required. Subfamilies CYP736 and CYP749, absent in Arabidopsis but present in some other plants, were also found in *Andrographis paniculata*. CYP736A300 in oregano (*Origanum vulgare*; *Lamiaceae*) catalyzes the conversion of phenolic monoterpenes such as thymol and carvacrol [23]. CYP749A16 enhances cotton (*Gossypium hirsutum*) resistance to the sulfonylurea herbicide trifloxysulfuron-sodium [24]. The functions of CYP subfamilies across different plants show correlations [25]. These findings suggest that the CYP736 subfamily in *Andrographis paniculata* may play a role in phenolic monoterpene conversion. Additionally, its CYP749 subfamily could be associated with herbicide resistance.

Through phylogenetic analysis and conserved domain alignment, 65 candidate diterpenoid pathway CYPs (hereinafter called candidate CYPs) were identified in *Andrographis paniculata.* They were further analyzed at the protein, RNA, and DNA levels. Nearly 97% of the candidate CYPs had peptide chains around 500 aa, indicating structural and functional similarities. Previous research shows that GGPP is primarily synthesized via the plastidial MEP pathway [26]. In this study, 88% of the candidate CYPs were localized to the chloroplast, with 12% being membrane-associated. Most candidate CYPs (77%) were transmembrane proteins, with transmembrane domains near the N-terminus, consistent with previous findings [27]. Signal peptides, leader peptides, and signal sequences are responsible for protein sorting and translocation. Signal peptides facilitate co-translational translocation, sorting proteins into the endomembrane system. Leader peptides and signal sequences function in post-translational translocation. They are responsible for sorting proteins into specific organelles, including mitochondria, chloroplasts, and peroxisomes [28]. No reliable signal peptides were predicted for the candidate CYPs. This suggested that their transmembrane localization relied solely on post-translational leader peptides or signal sequences.

The motif distribution of candidate CYPs exhibited subfamily specificity. These CYPs were derived from the CYP71, CYP76, CYP88A, CYP701A, and CYP714 subfamilies. The CYP714 subfamily had the fewest motifs, while the CYP71 subfamily had the most, including CYP71B and CYP71D. The CYP71D subfamily has been reported to contribute to diterpenoid biosynthesis in tobacco (*Nicotiana tabacum*) [29], tomato (*Solanum lycopersicum*), and *Euphorbiaceae* plants [22]. *Lamiaceae* subfamilies CYP71AU [30], CYP71BE [31], CYP71D [32], and *Lamiaceae*-specific subfamilies CYP76AH [33], CYP76BK [34], and CYP76AK [35] all participate in diterpenoid modification. CYP76 is closely associated with the synthesis of specialized labdane-type diterpenoids [36]. In *Andrographis paniculata* (*Acanthaceae*), the function of candidate CYP XP_051150854 was annotated as CYP76AH, suggesting similar diterpenoid modification pathways across different families within *Lamiales*. It is essential to conduct an in-depth analysis of XP_051150854’s role in the labdane-type diterpenoid andrographolide metabolic process. In contrast, the CYP71AU and CYP71BE subfamilies, found in both *Andrographis paniculata and Asteraceae* plants, have fewer specific expansion members [22]. Their broader roles in diterpenoid biosynthesis remain uncertain. The CYP99A subfamily (clustered with CYP71B in this study) appears more inclined towards modifying *Poaceae* diterpenoids [37]. In addition, the oxidation of *ent*-kaurene is critical for hormone synthesis in terrestrial plants. Most members of the CYP701A subfamily are functioning as *ent*-kaurene oxidases [38]. The ring-contraction reaction catalyzed by the CYP88A subfamily is essential for GA synthesis in vascular plants [39]. Members of the CYP714 subfamily deactivate GAs by catalyzing their hydroxylation. This results in dwarf and stress-resistant phenotypes in plants overexpressing *CYP714* [40].

The promoter *cis*-acting elements of the candidate *CYPs* were broadly connected with processes involving GAs, MeJA, auxin, ABA, and abiotic stress. This indicated that they were key factors in growth, development, and chemical defense. The extensive CYP71 subfamily not only modifies diterpenoids but also includes members involved in the modification of other terpenoids or flavonoids [41]. The promoters of two *CYP71* subfamily candidate *CYPs* (protein IDs: XP_051133617 and XP_051146596) contained flavonoid biosynthesis elements. Gene structure analysis revealed that one candidate *CYP* (protein ID: XP_051152769) contained an exceptionally long intron exceeding 40,000 bp. It harbored essential elements influencing transcription and protein function. These included light-responsive, hormone (ABA and GA) signaling, bZIP and WRKY binding sites, and abiotic stress-responsive elements. Based on the arrangement of elements within the core and proximal promoters, 10 representative CYPs from each subfamily were selected. The spatial structures of proteins from members of three GA pathway subfamilies were conserved. Additionally, the structures of members in the CYP71B, CYP76AH, and CYP99 subfamilies within the CYP71 family were similar. But they differed from the structure of the CYP71D subfamily member XP_051137327. This reflects the considerable functional divergence within the CYP71D subfamily [32].

LTs restrict the yield of *Andrographis paniculata*. In this study, prolonged LT stress significantly reduced andrographolide levels in the leaves. LT may limit plant growth by affecting the activity of diterpenoid pathway CYPs. It could also reduce andrographolide synthase activity or accelerate andrographolide degradation. Among the 10 representative *CYPs*, three [protein IDs: XP_051123242 (CYP88A, KAO), XP_051135400 (CYP701A, KO), and XP_051137327 (CYP71D)] responded significantly to LT stress. Studies have shown that GAs serve as a crucial factor in enhancing plant LT tolerance [42]. Treatment with 0.5 mmol/L GAs effectively maintains cell wall integrity in tomato fruit, improving postharvest LT resistance [43]. *KAO* and *KO* in the GA biosynthesis pathway exhibited upregulation under mild LT stress. This was followed by downregulation during moderate and severe stress. In contrast, the expression of *CYP71D* was consistently downregulated under LT treatment. These results indicated that *KAO* and *KO* functioned as positive regulators in the early LT response. *CYP71D* acted as a negative response factor throughout all stages of LT stress. Meanwhile, under LT conditions, KAO and KO interacted with each other, other proteins in the GA pathway, and a GA20OX2-like protein. However, *GA20OX2-like* showed minimal accumulation under LT treatment, with no clear trend relative to *KAO* and *KO*. This suggested that KAO, KO, and GA20OX2-like might have formed a trimeric complex under LT stress, but the role of GA20OX2-like was likely non-essential. KAO and KO collaborated to assist *Andrographis paniculata* in responding to early LT stress within the GA biosynthesis pathway. However, both might have had only a modest effect on rescuing andrographolide accumulation. By the later stages of LT stress, their transcript accumulation declined. The plant’s biological functions could have been severely impaired, leading to a significant decrease in andrographolide content. CYP71D interacted with 37 proteins, 95% of which were CYPs, with 46% belonging to the diterpenoid pathway CYPs identified in this study. This suggested that CYP71D might function as an auxiliary protein for CYPs. LT stress consistently suppressed its transcriptional activity. It might have been directly or indirectly involved in the synthesis of andrographolide under LT stress. In conclusion, plant CYPs make critical contributions to the chemical diversity of diterpenoids [44]. The identification and characterization of these CYPs in *Andrographis paniculata* can improve its environmental adaptability and enhance the pharmaceutical applications of andrographolide. Refs. [45,46] are cited in Appendix A.

## 4. Materials and Methods

### 4.1. Identification of Total CYPs in Andrographis Paniculata

The *Andrographis paniculata* genome (GenBank: GCA_009805555.1) and protein sequences were downloaded from NCBI (https://www.ncbi.nlm.nih.gov/). A Cytochrome P450 protein HMM model (PF00067) from Pfam 37.0 (http://pfam.xfam.org/, accessed on 5 December 2023) was used to identify CYPs via TBtools−II v2.067 (TBtools) Simple HMM search. Duplicate proteins were removed. Confirmed CYP sequences underwent NCBI Batch CD-search, NCBI BlastP (https://blast.ncbi.nlm.nih.gov/Blast.cgi?PROGRAM=blastp&PAGE_TYPE=BlastSearch&LINK_LOC=blasthome, accessed on 6 December 2023), and InterPro 98.0 database checks.

### 4.2. Construction of Phylogenetic Trees

CYPs from the diterpene pathway of terrestrial plants, excluding those belonging to gymnosperm-specific subfamilies, were selected as per reference [8], detailed in Appendix A. Arabidopsis CYPs were sourced from TAIR (https://www.arabidopsis.org/, accessed on 4 January 2024). CYPs with missing sequences, duplicate sequences, and those duplicating terrestrial plant diterpene pathway CYPs were removed. Using MEGA 11, the sequences of *Andrographis paniculata* CYPs, terrestrial plant diterpene pathway CYPs, and Arabidopsis CYPs were aligned with MUSCLE. Two overly short sequences (Arabidopsis AT5G35920 and *Andrographis paniculata* XP_051128743) were removed. The remaining CYPs were submitted to PhyloSuite_v1.2.3 for Gblock analysis to extract conserved protein sequences. A phylogenetic tree of conserved sequences was constructed using TBtools’ One Step Build an ML Tree. The *Andrographis paniculata* CYPs that clustered with terrestrial plant diterpene pathway CYPs were selected as candidates. A phylogenetic tree of full-length protein sequences was constructed for these candidates and terrestrial plant diterpene pathway CYPs. The standalone phylogenetic analysis of candidate diterpenoid pathway CYPs in *Andrographis paniculata* also utilized full-length protein sequences. Final trees were refined using iTOL v6 (https://itol.embl.de/, accessed on 12 January 2024) and Adobe Illustrator 2020 (AI).

### 4.3. Identification of Conserved Domains in Andrographis Paniculata CYPs

Conserved domains of plant CYPs include the I-helix (A/G-G-X-E/D-T-T/S), K-helix [K-E-T-L-R (E and R conserved) and P-E-R-F (P and R conserved)], and FXXGXRXCXG. FXXGXRXCXG is a characteristic signature for plant CYPs. The *Andrographis paniculata* CYP protein sequences were submitted to WebLogo (https://weblogo.berkeley.edu/logo.cgi, accessed on 3 February 2024). The analysis showed that all CYPs possessed the I-helix and K-helix domains. But, some lacked FXXGXRXCXG. This was possibly due to the poor genome sequencing or assembly quality of *Andrographis paniculata*. Using DNAMAN v9, the protein sequences of *Andrographis paniculata* CYPs were aligned, and 22 CYPs that lacked a complete FXXGXRXCXG were removed. The remaining full-length sequences were resubmitted to WebLogo. The generated images were enhanced using AI.

### 4.4. Analysis of Gene Structure

The full-length protein sequences of candidate CYPs were aligned using MUSCLE. They were submitted to TBtools One Step Build an ML Tree, saving the Newick data. Motifs were analyzed on MEME (https://meme-suite.org/meme/tools/meme, accessed on 10 February 2024) with classic mode. The upstream 2500 bp from the start codon was extracted as a promoter. The promoters were submitted to PlantCare (http://bioinformatics.psb.ugent.be/webtools/plantcare/html/, accessed on 16 February 2024) to screen for *cis*-acting elements. Gene lengths, positions of exons, introns, and untranslated regions were extracted. The Newick data, *cis*-acting element data, and gene component data were submitted to TBtools Gene Structure View (Advanced). AI was used to plot the gene structure diagrams. Functional elements of the unusually long intron were analyzed using PLACE (https://www.dna.affrc.go.jp/PLACE/?action=newplace, accessed on 2 May 2024) [21].

### 4.5. Analysis of Protein Characteristics

Predictive analyses included molecular mass (Expasy, https://www.expasy.org/, accessed on 13 May 2024), subcellular localization (WoLF PSORT, https://wolfpsort.hgc.jp/, accessed on 13 May 2024), transmembrane domains (TMHMM-2.0, https://services.healthtech.dtu.dk/services/TMHMM-2.0/, accessed on 17 May 2024), signal peptides (SignalP 5.0, https://services.healthtech.dtu.dk/services/SignalP-5.0/, accessed on 17 May 2024), 2D structure (PSIPRED, http://bioinf.cs.ucl.ac.uk/psipred/), and 3D structure (SWISS-MODEL, https://swissmodel.expasy.org/, accessed on 17 May 2024). Protein sequences were submitted to the STRING database (https://cn.string-db.org/, accessed on 17 May 2024). The interaction networks were constructed using Cytoscape. Interactions that had a combined score greater than 500 were retained. The image was annotated with AI.

### 4.6. Quantification of Andrographolide Content

The seeds of *Andrographis paniculata*, which are already widely cultivated, were sourced from Raoping County, Chaozhou City, Guangdong Province. They were cultivated at the Guangdong Provincial Key Laboratory of Crop Genetic Improvement, Crop Research Institute, Guangdong Academy of Agricultural Sciences.

#### 4.6.1. Growth Conditions

The seeds were sterilized with 75% ethanol and 10% sodium hypochlorite, then sown in sterile peat soil and incubated at 28 °C with long-day conditions (16 h light/8 h dark) and 75% humidity. The plants were grown in pots, with Hoagland nutrient solution added every 3–5 days to maintain soil moisture. After 60 days, the seedlings were about 20 cm tall and had four pairs of fully expanded leaves, excluding the cotyledons. The roots, stems, and third leaves from the top were harvested for andrographolide content analysis. Additional seedlings were subjected to LT treatment.

#### 4.6.2. LT Treatment Conditions

The seedlings were maintained at 20 °C for 8 weeks. LT damage to the leaves was assessed by observing reddening. The third leaves from the top were collected for andrographolide analysis at 0 weeks (CK, green leaves) and 8 weeks (AR, all red leaves) of LT treatment. Three independent biological replicates were used for each condition, each from at least 2 seedlings.

#### 4.6.3. Detection of Andrographolide Using Liquid Chromatography–Tandem Mass Spectrometry

The andrographolide content was measured using liquid chromatography (Shim-pack UFLC SHIMADZU CBM30A, Shimadzu Corporation, Kyoto, Japan) with tandem mass spectrometry (Applied Biosystems 4500 QTRAP, Thermo Fisher Scientific, Waltham, Massachusetts, United States). The andrographolide standard has a CAS number of 5508-58-7 [47].

Chromatography was performed on a Waters ACQUITY UPLC HSS T3 C18 column (1.8 µm, 2.1 mm × 100 mm). The mobile phase included ultrapure water with 0.04% acetic acid (Phase A) and acetonitrile with 0.04% acetic acid (Phase B). The gradient was as follows: 0 min A/B 95:5, 10 min A/B 5:95, 11 min A/B 5:95, 11.1 min A/B 95:5, and 14 min A/B 95:5. The flow rate was 0.35 mL/min, with a column temperature of 40 °C and an injection volume of 3 μL.

Mass spectrometry settings were the following: ESI ion source, 550 °C temperature, 5500 V ion source voltage, 35 psi curtain gas, CAD -2, and optimized DP and CE.

#### 4.6.4. Statistical Analysis

The mean and standard deviation (SD) of the three replicates were calculated. A Student’s *t*-test was conducted using SPSS 19.0 to determine significant differences at *p* < 0.05 (*) or *p* < 0.01 (**).

### 4.7. Gene Expression Analysis under LT Treatment

The third leaves from the top were collected at 0 weeks (CK), 2 weeks [mild stress, slight red leaves, SR], 4 weeks [moderate stress, medium red leaves, MR], and 8 weeks (AR) of LT treatment. The collected leaves were rapidly frozen in liquid nitrogen and stored at −80 °C. Three independent biological replicates were taken for each treatment, with each replicate consisting of leaves from at least two seedlings.

The total RNA extraction and transcriptome sequencing were conducted by Guangzhou Gene Denovo Biotechnology Co., Ltd. The FPKM values of the target genes in CK and treatment groups were log_2_-transformed (Appendix A). They were submitted to TBtools HeatMap for generating heat maps of gene expression levels. The sequence data reported in this study were archived in the Sequence Read Archive (SRA) with the accession number PRJNA1167344.

The images were assembled with AI.

## 5. Conclusions

We systematically identified and classified 345 CYPs in *Andrographis paniculata*. Of these, 328 belonged to 42 known subfamilies, while 17 were possibly unique to *Andrographis paniculata*. Subfamilies CYP736 and CYP749, absent in Arabidopsis but present in some other plants, were also identified. Phylogenetic analysis and domain alignment revealed 65 candidate CYPs involved in diterpenoid pathways. These candidates were mostly transmembrane proteins localized to the chloroplast and had similar molecular sizes. Their motif distribution showed subfamily specificity. Promoter analysis indicated that these candidates were key factors in growth, development, and chemical defense. Based on the characteristics of *cis*-acting element arrangement and subfamily classification, 10 representative candidate CYPs were selected. The expression patterns under LT stress revealed that *KAO* (protein ID: XP_051123242) and *KO* (protein ID: XP_051135400) acted as positive regulators in the early LT response. *CYP71D* (protein ID: XP_051137327) functioned as a negative response factor throughout LT stress. CYP71D interacted with 37 proteins, 95% of which were CYPs, including 46% from the diterpenoid pathway. KAO and KO were found to interact with each other and with other proteins in the GA pathway, suggesting a coordinated role in response to LT stress. These findings enhance our understanding of the functional diversity of CYPs in *Andrographis paniculata*, particularly their roles in diterpenoid metabolism and LT responses. This knowledge provides a foundation for future studies aimed at improving the environmental adaptability of *Andrographis paniculata* and enhancing the pharmaceutical applications of andrographolide. The candidates supply direction for research on andrographolide and GAs in molecular biology, synthetic biology, and related fields.

## Figures and Tables

**Figure 1 ijms-25-10741-f001:**
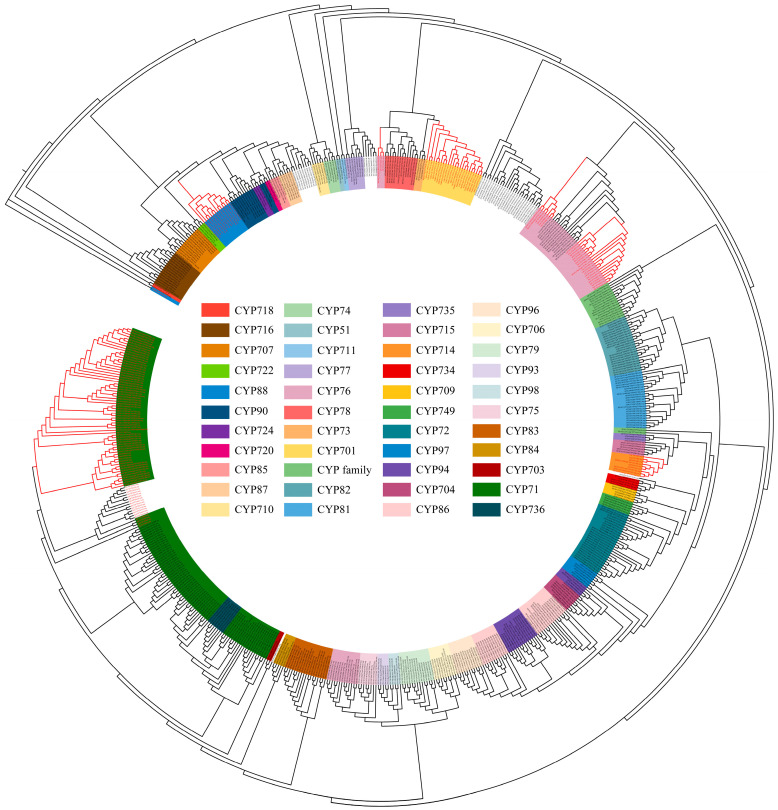
Phylogenetic tree of conserved CYP protein sequences in *Arabidopsis* (total CYPs), *Andrographis paniculata* (total CYPs), and terrestrial plants (diterpenoid pathway CYPs) Note: red text—diterpenoid pathway CYPs in *Andrographis paniculata* and terrestrial plants; red branches—clades containing *Andrographis paniculata* CYPs within the diterpenoid pathway.

**Figure 2 ijms-25-10741-f002:**
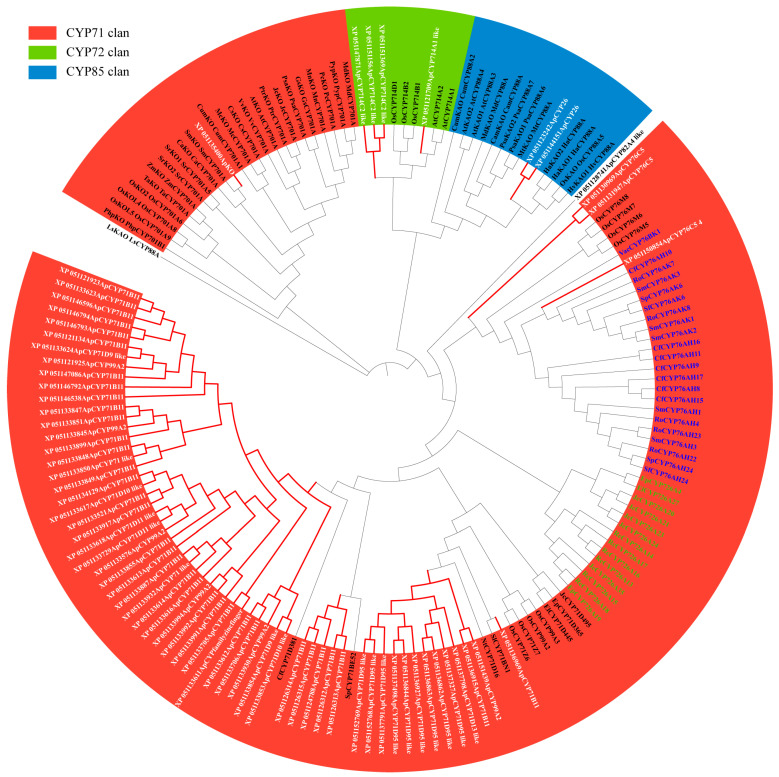
Phylogenetic tree of full-length CYP protein sequences from diterpene pathways in *Andrographis paniculata* and terrestrial plants Note: green text—*Euphorbiaceae*-specific expanded subfamily CYP726A; blue text—*Lamiaceae*-specific expanded subfamily CYP76AH, CYP76AK, and CYP76BK; red branches (white text)—*Andrographis paniculata* candidate diterpenoid pathway CYPs; black branches—terrestrial plants diterpenoid pathway CYPs.

**Figure 3 ijms-25-10741-f003:**
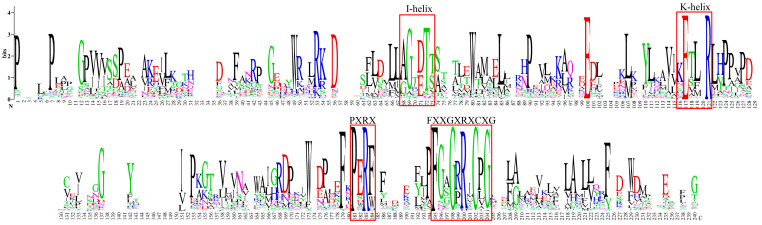
Conserved domains of 323 *Andrographis paniculata* CYPs. Note: red squares—conserved domains.

**Figure 4 ijms-25-10741-f004:**
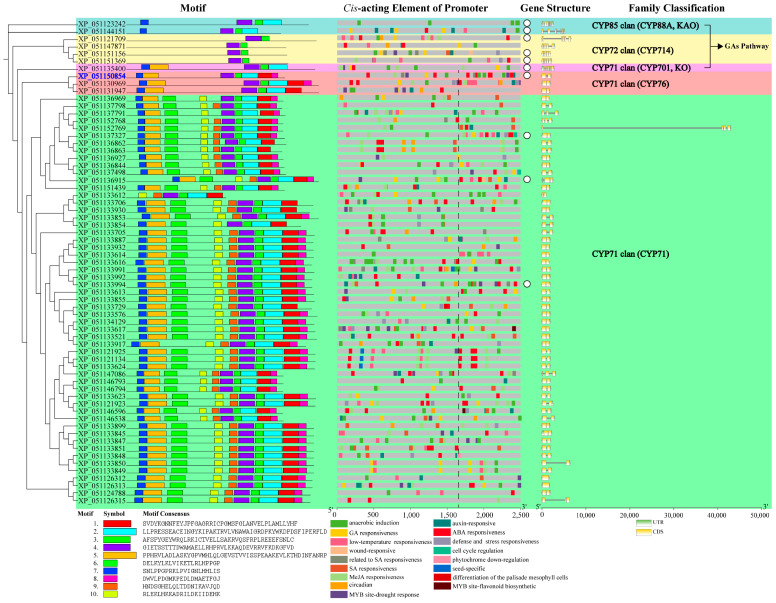
The distribution of protein motifs, promoter *cis*-acting elements, and gene structures of candidate diterpenoid pathway CYPs Note: blue ID—candidate CYP clustered in the *Lamiaceae*-specific expanded CYP subfamily; promoter region to the right of the dashed line—core and proximal promoters; white circles—10 representative candidate CYPs.

**Figure 5 ijms-25-10741-f005:**
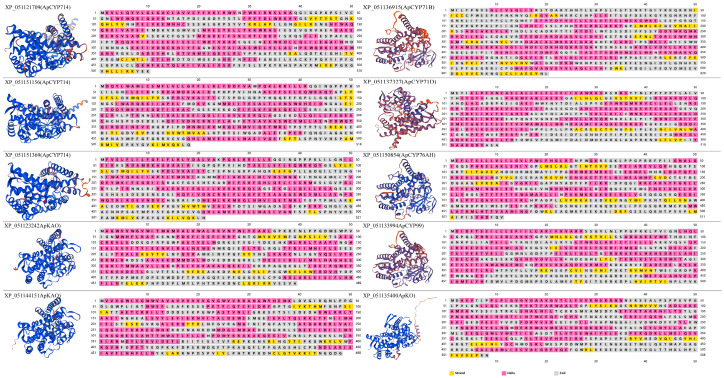
3D and 2D structures of representative candidate CYPs.

**Figure 6 ijms-25-10741-f006:**
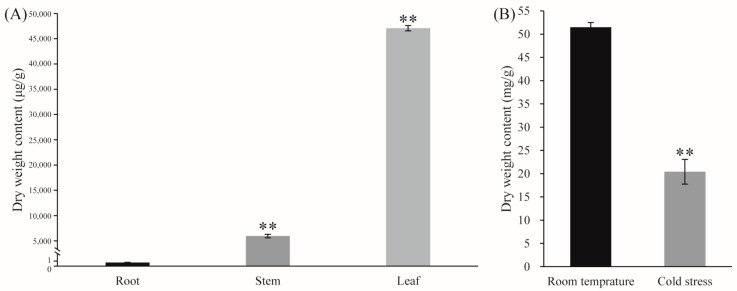
(**A**) Tissue-specific accumulation of andrographolide; (**B**) andrographolide content in leaves under LT treatment Note: The data were represented as the averages of three independent biological experiments ± SD, and asterisks indicated a significant difference compared to the corresponding controls.

**Figure 7 ijms-25-10741-f007:**
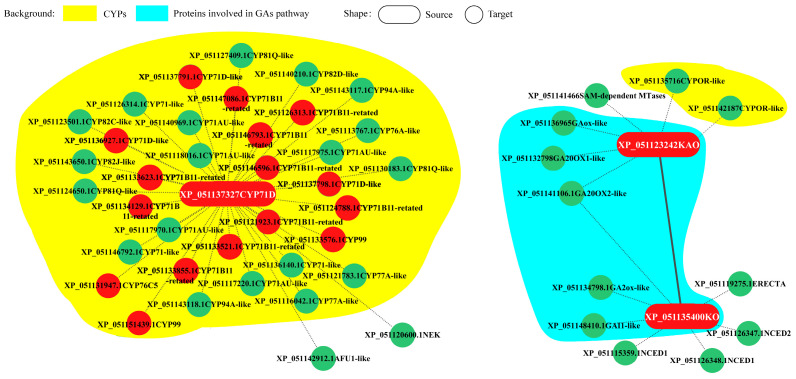
Protein interaction of KAO, KO, and CYP71D Note: red—diterpenoid pathway CYPs; green—other interacting proteins.

**Figure 8 ijms-25-10741-f008:**
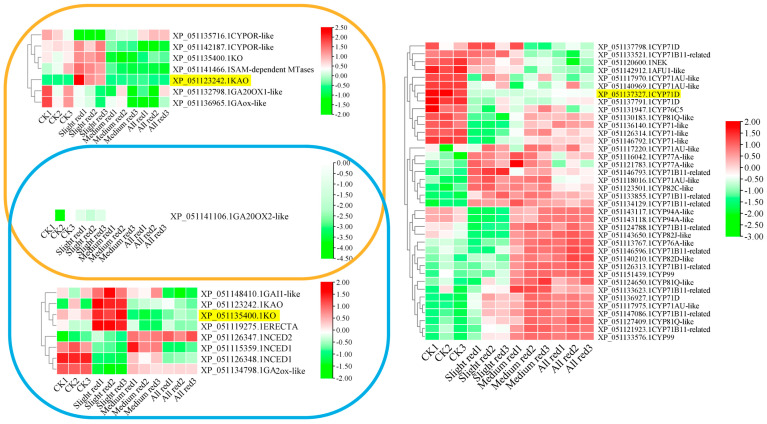
LT−related expression profiles of *KAO*, *KO*, and *CYP71D* and transcripts of their interacting proteins. Note: yellow IDs—protein IDs of representative *CYPs*.

**Table 1 ijms-25-10741-t001:** Distribution of CYP subfamilies.

No.	CYP Subfamily	Number of *Andrographis Paniculata* CYPs	Number of *Arabidopsis Thaliana* CYPs	Number of Terrestrial Plants Diterpenoid Pathway CYPs
1	51	1	1	0
**2**	**71**	**80**	**48**	**11**
3	72	20	8	0
4	73	2	1	0
5	74	3	2	0
6	75	7	1	0
**7**	**76**	**23**	**9**	**26**
8	77	3	4	0
9	78	8	6	0
10	79	8	6	0
11	81	8	16	0
12	82	19	5	0
13	83	15	3	0
14	84	4	1	0
15	85	3	1	0
16	86	17	10	0
17	87	5	1	0
**18**	**88**	**2**	**2**	**12**
19	89	0	6	0
20	90	9	4	0
21	93	4	1	0
22	94	12	6	0
23	96	0	12	0
24	97	4	3	0
25	98	2	3	0
**26**	**701**	**1**	**1**	**23**
27	702	0	6	0
28	703	1	1	0
29	704	5	3	0
30	705	0	25	0
31	706	2	7	0
32	707	10	4	0
33	708	0	3	0
34	709	2	3	0
35	710	1	4	0
36	711	1	1	0
37	712	0	2	0
**38**	**714**	**4**	**2**	**5**
39	715	5	1	0
40	716	13	2	0
41	718	1	1	0
42	720	1	1	0
43	721	0	1	0
44	722	2	1	0
45	724	2	1	0
**46**	**726**	**0**	**0**	**14**
47	734	4	1	0
48	735	1	2	0
49	736	8	0	0
50	749	5	0	0
51	CYP family	17	0	0
	Total	345	233	91

Note: bold subfamilies—diterpenoid pathway subfamilies in terrestrial plants; red subfamilies—subfamilies clustered exclusively with Andrographis paniculata. Total Andrographis paniculata CYPs: 345 (XP_051128743 excluded due to misalignment). Total Arabidopsis thaliana CYPs: 233 [includes 5 additional CYPs in the terrestrial plant diterpene pathway; AT1G17060 (CYP72C1) excluded as it did not cluster with the CYP72 subfamily; AT5G35920 excluded due to misalignment]. Total terrestrial plant diterpenoid pathway CYPs: 91 [LsKAO (LsCYP88A) excluded as it did not cluster with other KAO (CYP88) proteins].

**Table 2 ijms-25-10741-t002:** Promoter *cis*-elements of candidate *CYPs* involved in the diterpenoid pathway in *Andrographis paniculata*.

Subfamily	Gene Number	Number of *Cis*-Elements											
		ABA	Anaerobic	Auxin	Stress	GAs	LT	MeJA	Drought	Circadian	Salicylic Acid	Cell Differentiation	Flavonoid Biosynthesis
CYP88A	2	1	6	4	1	3	1	0	0	0	0	0	0
CYP701A	1	1	3	0	0	2	0	3	1	0	0	0	0
CYP714s	4	5	6	2	3	5	8	4	2	2	4	0	0
CYP76s	3	17	7	3	2	3	6	5	7	3	2	1	0
CYP71s	55	120	78	30	25	40	53	97	29	25	55	4	2
Total	65	144	100	39	31	53	68	109	39	30	61	5	2

## Data Availability

Data are contained within the article.

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
