# Peer review of "Genome-Wide Identification and Characterization of Diterpenoid Pathway CYPs in Andrographis paniculata and Analysis of Their Expression Patterns under Low Temperature Stress"

_ijms, 2024, doi:10.3390/ijms251910741_

Round 1
Reviewer 1 Report
Comments and Suggestions for Authors
The manuscript contains bioinformatic analysis dedicated to CYP genes in Andrographis paniculata together with expression changes in response to low temperatures.
Several issues needs to be explained:
LT treatment was in 20ºC – it is hard to accept this temperature as low – how it was selected, optimized – what are normal temperatures during growth season of Andrographis paniculata.
As MVA and MEP pathways are generally well conserved in plants It would be interested what enzymes are involved in synthesis of target metabolite – andrographolide and some diagram with reduced MVA and MEP part would be helpful in introduction.
The 668 proteins used in phylogenetic analysis remain anonymous – and supplementary data with these reference data are necessary.
Source data with raw reads of RNA seq are not provided. Authors studied expression in plants after several days spaces together with andrographolide content and results are not presented – it would be important to get confirmation if expression level of genes active in andrographolide content is reduced together with andrographolide decrease and what is the further fate of this compound (what genes are activated.
RNAseq analyses are too selective – some data on the importance of CYP enzymes in total reaction to temperature of 20ºC is necessary to have a view on the role of this process in general response to LT. We do not know coverage of RNA-seq analysis.
Final conclusion is unclear – how these results enhance the pharmaceutical applications of andrographolide? Optimally, isoforms active in synthesis of andrographolide should be indicated.
For breeding purposes, data on CYP enzymes are valuable but mechanism of adaptation is studied on single not defined/named genotype – data on plant variety name needs to be supplemented. For this experimental design information if the studied species is self-pollinated (so genetic diversity of plants is reduced) or outcrosser – than genetic variation could interfere with expression patterns. It is also hard to extrapolate information from single genotype/genepool on whole species – therefore for LT tolerance studies extremely sensitive and tolerant genotypes should be selected – we do not know status of genotype under investigation.
Reviewer 2 Report
Comments and Suggestions for Authors
This manuscript conducted the genome-wide identification and characterization of diterpenoid pathway CYPs in Andrographis paniculata and analyzed their expression patterns responding to low temperature stress. This research work is very meaningful and valuable. But the manuscript need to be revised and improved. I have some comments and suggestions that require the author's response and improvement.
1. I suggest that, Tables 2 and 4 can be listed as supplementary tables.
2. The font in the pictures are too small to see clearly. Please enlarge the font of words in Figures 7 and 8.
3. In the Materials and Methods section, the description of plant materials is not clear enough, such as how big the plant is? Please show the phenotypes of plant seedlings under LT stress.
4. Additionally, do the control materials need to be set at the same time period as the stress treatment for 2, 4, and 8 weeks?
5. I suggest that, qPCR experiments should be conducted to validate the results of transcriptome analysis.
Reviewer 3 Report
Comments and Suggestions for Authors
In the manuscript “Genome-Wide Identification and Characterization of Diterpenoid Pathway CYPs in Andrographis paniculata and Analysis of Their Expression Patterns under Low Temperature Stress” Mingyang Sun and co-authors assessed Cytochrome P450 monooxygenases family in Andrographis patniculata .Based on the plant CYP HMM model, NCBI, and Interpro annotations, 346 CYPs in Andrographis paniculata were identified.Under Low Temperature stress, significant expression changes were observed in three representative CYPs: CYP71D, KAO, and KO. The text are described with enough detail to allow others to replicate and build on published results. The paper provides a concise and precise description of the results. I suggest to deeply revise the text, there are several formatting errors.
1. In the abstract, it can be written how many CYPs gene members there are in Andrographis paniculata, which can be divided into several subfamilies.
2. KAO and KO should use their full names in the abstract.
3. Table1 incomplete display.
4. The tables in this article are quite long. I suggest that some tables can be attached as supplementary materials, especially Table 5
5. In discussion section, “They were further analyzed at” in line 83 page 21, italics should not be used.
6. Discussion on the relationship between andrographolide and low temperature should be added.
7. “4.6. Quantification of Andrographolide Content” this section can be briefly described.
Author Response
Manuscript title: Genome-Wide Identification and Characterization of Diterpenoid Pathway CYPs in Andrographis paniculata and Analysis of Their Expression Patterns under Low Temperature Stress
Manuscript ID: ijms-3207165
Journal submitted: International Journal of Molecular Sciences
Date of comments received: 15 Sep 2024
Responses to Reviewer 3 Comments
Summary: In the manuscript “Genome-Wide Identification and Characterization of Diterpenoid Pathway CYPs in Andrographis paniculata and Analysis of Their Expression Patterns under Low Temperature Stress” Mingyang Sun and co-authors assessed Cytochrome P450 monooxygenases family in Andrographis patniculata. Based on the plant CYP HMM model, NCBI, and Interpro annotations, 346 CYPs in Andrographis paniculata were identified. Under Low Temperature stress, significant expression changes were observed in three representative CYPs: CYP71D, KAO, and KO. The text is described with enough detail to allow others to replicate and build on published results. The paper provides a concise and precise description of the results. I suggest to deeply revise the text, there are several formatting errors.
Response: Thank you for your thorough review and valuable suggestions.
Comment [1]: In the abstract, it can be written how many CYPs gene members there are in Andrographis paniculata, which can be divided into several subfamilies.
Response [1]: The relevant content has been added to the abstract.
“346 CYPs were discovered in Andrographis paniculata. Among them, 328 CYPs belonged to 42 known subfamilies. The remaining 17 CYPs might have represented novel subfamilies unique to this species.” (Revised manuscript, page 1, lines 20-23).
Comment [2]: KAO and KO should use their full names in the abstract.
Response [2]: The full names of KAO and KO have been added to the abstract.
“CYP71D, ent-kaurenoic acid oxidase (KAO), and ent-kaurene oxidase (KO).” (Revised manuscript, page 1, line 36).
Comment [3]: Table1 incomplete display.
Response [3]: The layout of Table 1 has been modified (Revised manuscript, pages 5-6).
Comment [4]: The tables in this article are quite long. I suggest that some tables can be attached as supplementary materials, especially Table 5
Response [4]: Tables 2, 4, and 5 have been included as supplementary data (Table S2, S3, and S4).
Comment [5]: In discussion section, “They were further analyzed at” in line 83 page 21, italics should not be used.
Response [5]: The font has been adjusted (Revised manuscript, page 23, line 83).
Comment [6]: Discussion on the relationship between andrographolide and low temperature should be added.
Response [6]: The relevant content has been added to the discussion.
The sentences are” In this study, prolonged LT stress significantly reduced andrographolide levels in the leaves. LT may limit plant growth by affecting the activity of diterpenoid pathway CYPs. It could also reduce andrographolide synthase activity or accelerate andrographolide degradation.” (Revised manuscript, page 24, lines 157-162); “KAO and KO collaborated to assist Andrographis paniculata in responding to early LT stress within the GAs biosynthesis pathway. However, both might have had only a modest effect on rescuing andrographolide accumulation. By the later stages of LT stress, their transcript accumulation declined. The plant's biological functions could have been severely impaired, leading to a significant decrease in andrographolide content.” (Revised manuscript, page 24, lines 182-186; and page 25, lines 187-189); and “LT stress consistently suppressed its transcriptional activity. It might have been directly or indirectly involved in the synthesis of andrographolide under LT stress.” (Revised manuscript, page 25, lines 192-195).
Comment [7]: “4.6. Quantification of Andrographolide Content” this section can be briefly described.
Response [7]: Section 4.6. has been briefly described. The total word count of sections 4.6.1 to 4.6.4 has been reduced from 400 to 344. In section 4.6.1, a description of the experimental seedlings' phenotypes has been added (The content highlighted below).
The sentences are ”4.6.1. Growth Conditions
The seeds were sterilized with 75% ethanol and 10% sodium hypochlorite, then sown in sterile peat soil and incubated at 28°C with long-day conditions (16 hours light/8 hours dark) and 75% humidity. The plants were grown in pots, with Hoagland nutrient solution added every 3-5 days to maintain soil moisture. After 60 days, the seedlings were about 20 cm tall and had four pairs of fully expanded leaves, excluding the cotyledons. The roots, stems, and third leaves from the top were harvested for andrographolide content analysis. Additional seedlings were subjected to LT treatment.
4.6.2. LT Treatment Conditions
The seedlings were maintained at 20°C for 8 weeks. LT damage to the leaves was assessed by observing reddening. The third leaves from the top were collected for andrographolide analysis at 0 weeks (CK, green leaves) and 8 weeks (AR, all red leaves) of LT treatment. Three independent biological replicates were used for each condition, each from at least 2 seedlings.
4.6.3. Detection of Andrographolide by Liquid chromatography-tandem mass spectrometry
The andrographolide content was measured using liquid chromatography (Shim-pack UFLC SHIMADZU CBM30A) with tandem mass spectrometry (Applied Biosystems 4500 QTRAP). The andrographolide standard has a CAS number of 5508-58-7.
Chromatography was performed on a Waters ACQUITY UPLC HSS T3 C18 column (1.8 µm, 2.1 mm x 100 mm). The mobile phase included ultrapure water with 0.04% acetic acid (Phase A) and acetonitrile with 0.04% acetic acid (Phase B). The gradient was as follows: 0 min A/B 95:5, 10 min A/B 5:95, 11 min A/B 5:95, 11.1 min A/B 95:5, and 14 min A/B 95:5. The flow rate was 0.35 ml/min, with a column temperature of 40°C and an injection volume of 3 μl.
Mass spectrometry settings were: ESI ion source, 550°C temperature, 5500 V ion source voltage, 35 psi curtain gas, CAD -2, and optimized DP and CE.
4.6.4. Statistical Analysis
The mean and standard deviation (SD) of the three replicates were calculated. Student’s t-test was conducted using SPSS 19.0 to determine significant differences at p < 0.05 (*) or p < 0.01 (**).” (Revised manuscript, page 30, lines 42-48; and page 31, lines 49-97).
Round 2
Reviewer 1 Report
Comments and Suggestions for Authors
I accept the most of explanations and thank authors for providing supplementary data. I do not insist on the scheme presented in paper 3 while the citation of the article https://doi.org/10.1111/tpj.14162 regarding the later stages of transformation is missing. As for comment #4 it is a misunderstanding - I meant that in most of articles raw sequencing data sequencing data are deposited in some globally accessible scientific database. Without SRA or raw sequence data I cannot recommend acceptance of this manuscript.
Reviewer 2 Report
Comments and Suggestions for Authors
I am satisfied with the author's revisions made to the manuscript and their responses, and I recommend to be accepted for potential publication.
Author Response
Comment: I am satisfied with the author's revisions made to the manuscript and their responses, and I recommend to be accepted for potential publication.
Response: Thank you for your acknowledgment of our work. We sincerely appreciate it
Round 3
Reviewer 1 Report
Comments and Suggestions for Authors
Thank you. As soon as manuscript will be supplemented with valid NCBI number leading to raw sequence data it can be published without further delay.
Author Response
Comment: Thank you. As soon as manuscript will be supplemented with valid NCBI number leading to raw sequence data it can be published without further delay.
Response: We have added the accession number of the raw data in the revised manuscript (Page 23, lines 82-84). As the same raw data is also used in another unpublished study of ours, we have chosen to release it after one year.